# Evolution of Preset Void and Damage Characteristics in Aluminum during Shock Compression and Release

**DOI:** 10.3390/nano12111853

**Published:** 2022-05-28

**Authors:** Ya-Ting Wan, Jian-Li Shao, Guang-Ze Yu, Er-Fu Guo, Hua Shu, Xiu-Guang Huang

**Affiliations:** 1Shanghai Institute of Laser Plasma, China Academy of Engineering Physics, Shanghai 201800, China; ytwan@zju.edu.cn (Y.-T.W.); guangze.yu@foxmail.com (G.-Z.Y.); guoerfu@hotmail.com (E.-F.G.); 2State Key Laboratory of Explosion Science and Technology, Beijing Institute of Technology, Beijing 100081, China; 3Explosion Protection and Emergency Disposal Technology Engineering Research Center of the Ministry of Education, Beijing 100039, China

**Keywords:** molecular dynamics, shock pressure, void evolution, strain rate

## Abstract

It is well known that initial defects play an essential role in the dynamic failure of materials. In practice, dynamic tension is often realized by release of compression waves. In this work, we consider void-included single-crystal aluminum and investigate the damage characteristics under different shock compression and release based on direct atomistic simulations. Elastic deformation, limited growth and closure of voids, and the typical spall and new nucleation of voids were all observed. In the case of elastic deformation, we observed the oscillatory change of void volume under multiple compression and tension. With the increase of impact velocity, the void volume reduced oscillations to the point of disappearance with apparent strain localization and local plastic deformation. The incomplete or complete collapsed void became the priority of damage growth under tension. An increase in sample length promoted the continuous growth of preset void and the occurrence of fracture. Of course, on the release of strong shock, homogeneous nucleation of voids covered the initial void, leading to a wider range of damaged zones. Finally, the effect of the preset void on the spall strength was presented for different shock pressures and strain rates.

## 1. Introduction

In real metals, there are different types of defects, such as impurities, voids, grain boundaries, and dislocations [1,2], which are known as the original sites of damage development and can also change the spall strength and other mechanical properties of materials. As we all know, the essence of ductile metallic material fracture is a typical multiscale behavior, including the nucleation, growth, and coalescence of nanovoids [3,4,5,6]. The application of pump–probe measurements together with cluster techniques may classify the quality of the samples depending on the tiny differences of the optical readout upon the pump–probe measurements [7,8,9,10].

However, it is difficult to track the real-time evolution of voids, especially submicron voids, using conventional skills. Generally, microscale modeling combined with experimental observation is very important to understand the mechanical properties of materials [11]. Here, molecular dynamics (MD) simulation was chosen, which mainly relies on Newtonian mechanics to simulate the motion of molecular systems and further calculate the thermodynamic quantities and other macroscopic properties of the system. Nowadays, it has been widely used in the equilibrium and nonequilibrium calculation of materials at submicron scales [12].

For the microdefect deformation and damage mechanism, several studies were conducted by MD simulations to reveal that the preset void affected the elastic modulus and yield strength of materials. At the same time, the effects of porosity, temperature, strain rate, and grain size on the evolution of voids and the overall mechanical properties of materials were discussed [13,14]. Li et al. analyzed the uniaxial straining deformation behaviors of both single-crystal and nanotwinned copper materials embedded with a pre-existing spheroidal void [15]. The coupling effects among twin boundaries, spheroidal void aspect ratios and orientation on unidirectional elastic–plastic behavior, were systematically studied by the elastic modulus, yield stress, first peak stress, and plasticity index. Zhao et al. studied the effects of the initial radius and initial spacing of voids on material evolution [16]. Dislocation analysis showed that the dislocation emission on the void surface was an important behavior in the initial stage of void evolution. In addition, the effects of the length–width ratio, orientation, and position of spherical void on elastic–plastic behavior were also studied [17,18,19,20].

Nevertheless, the existing studies mainly focus on the evolution behavior of the preset void under single compression or tension and the evolution mechanism of the void in the whole process of shock compression and release needs to be further studied, especially in dynamic failure with a variation of pressure history and strain rate. In this work, we constructed a single-crystal aluminum model with one void in the center and explored the effect of preset void on the tensile failure performance of the material after shock compression. The release strain rate was changed by controlling the scale of the model in the shock direction, while the shock pressure was changed by controlling the impact velocity. Furthermore, by observing the microstructure evolution and analyzing the changes of temperature and stress, the processes of elastic deformation, limited growth and closure of voids, and typical spall and uniform nucleation were observed. In addition, the free surface velocity and the variation of the spall strength were presented.

## 2. Model and Simulation Details

A LAMMPS (large-scale atomic/molecular massively parallel simulator) was chosen in our simulations [21], which has been widely used to study the shock response of metallic materials. Our simulated sample is shown in Figure 1 with lengths Lx = 45a, Ly = 45a, and Lz = 120a/600a/1200a/2000a (identified by L1, L2, L3, and L4, respectively), where the x, y, and z axes present the [100], [010], and [001] directions, and a is the lattice constant. For L1, the model consists of 10^6^ atoms. Different sample sizes were used to produce different strain rates. In order to observe the evolution process of the void under the impact, the void was designed in the center of the model with a diameter of 2a, which is the exact position of the spall plane. The embedded-atom method (EAM) potential, developed by Zhakhovskii, was utilized here, as it has been applied in a wide range of pressure and can describe the spall of aluminum precisely [22]. Periodic boundary conditions were employed along the x and y directions to reduce the boundary effects, and for the z-direction, defined as the shock direction, free surface was employed for observing the material evolutions after impact. The motion equation of atoms was integrated by the velocity Verlet algorithm with a time step of 1 fs. Before compression, the sample was set under periodic boundary conditions in x, y, z axes and adjusted to the temperature of 300 K and zero pressure by the Nosé–Hoover method in isothermal–isobaric (npt) ensembles. After 5 ps relaxation, the model was in an equilibrium state, confirmed by temperature, volume, and energy balance. As for the implementation of the shock compression, the model was divided into two identical parts in the z-direction with opposite velocities ±v_0_ to generate shock waves by impact. Different release strain rates were obtained through sample size settings, and different shock pressure platforms were investigated by adjusting the impact velocity v_0_ (0.25, 0.5, 1, and 2 km/s), covering elastic deformation, dislocation emission, and complete collapse of the void. The peak shock stress was varied from 4 to 45 GPa, including the shock or release melting cases.

Various physical parameters and microscopic images were obtained in order to analyze the simulation results. The stress tensor was calculated according to the microscopic formula of Irving and Kirkwood [23], and the temperature was obtained after subtracting the velocity of the center-of-mass. As for the microstructure image, Ovito was utilized for post processing and observation [24]. In particular, the deformation of each atom was analyzed by its centrosymmetry parameter *p**_CSP_*, which is typically used to measure the local disorder around a single atom and identify whether the atom is in the ideal lattice or in the defect (such as a dislocation or stacking fault) or at a surface. *p**_CSP_* is defined as follows:$$p_{CSP}=\underset{i=1,6}{\sum }{\left|R_{i}+R_{i+6}\right|}^{2}$$
*R_i_* and *R_i_*_+6_ are the vectors corresponding to the six pairs of opposite nearest neighbors in the initial fcc lattice.

At the same time, the construct surface mesh was adopted to show the void distribution, facilitating our observation of void collapse process and nucleation during subsequent tension. Atomic displacement vectors were also calculated to display changes in the position of the atoms, and crystal structure types were counted by common neighbor analysis.

## 3. Results and Discussion

### 3.1. Effects of Shock Pressure and Strain Rate of Release on Microstructure Evolution

In the case of model L1, the strain rates were about 10^10^–10^11^ s^−1^. To be exact, the release strain rates were calculated by the volume change of the sample when the rarefaction waves were encountered. In order to investigate the effect of the strain rate on microstructure evolution and spall damage, we further increased the scales of the model along the impact direction. Under such circumstances, the strain rates ranged from 10^10^ to 10^8^ s^−1^. More specifically, the corresponding strain rates under different model sizes and shock pressures are shown in Table 1.

#### 3.1.1. Void Evolution under Elastic Compression and Release

Firstly, elastic deformation was observed for model L1 at the impact velocity of 0.25 km/s. In particular, the preset void expands and contracts with time and its volume change is approximately proportional to the sample volume change, with no noticeable effects on the subsequent recompression and stretching processes. The initial size of the sample before impact was 18.2 × 18.2 × 48.6 nm^3^, and the diameter of the void along the shock direction was 1.6 nm. At 4 ps, the shock wave reached the free surface (Figure 2a), and the sample was compressed to minimum, accompanied by the reduction of void size to 1.5 nm. The peak stress also appeared at this moment, about 4.5 GPa. At 12.5 ps, the sample was stretched to the maximum length, accompanied by the maximum void size of 1.8 nm with individual atom migration at the void boundary (Figure 2b). As shown in the sample volume profile, under the current simulation conditions, the period of elastic deformation was about 17 ps (Figure 3a). Therefore, at 21 ps, the sample returned to the state of compression limit, while the sample volume and void size were also consistent with that of 4 ps, and the atomic migration was restored.

The temperature change along the shock direction was also analyzed, which remained basically constant between 270 and 330 K with limited symmetrical fluctuations compared to the equilibrium temperature 300 K, due to the evolution of microscopic structure induced by impact (Figure 3b). The von Mises stress is a criterion with which to judge whether a material has plastic change, and its evolution in the preset void layer over time was counted. As shown in Figure 3a, the von Mises stress maintains periodicity, indicating the lack of plastic deformation. Similarly, the sample and void volumes changed periodically with the evolution. The difference is that changes in void volume as well as the von Mises stress were almost instantaneous, while the sample volume changed continuously over time (Figure 3c). More microcosmic deformation can be observed around the preset void by analyzing the atomic displacement vectors, and it can be concluded that locally, the collapse and stretching of the void was almost elastic with the migration of individual atoms in a certain direction, appearing inward or outward, respectively, over time (Figure 3c). The animation of corresponding void is provided in the Appendix A. The mechanical properties of the whole sample also reflect the complete elastic evolution process.

In models L2, L3, and L4 with the same impact velocity, the microstructure deformation, particle velocity, stress, and temperature profiles are very similar to those of L1 with v_0_ = 0.25 km/s. The models L2, L3, and L4 only exhibit elastic deformation with the evolution periods of 80, 120, and 273 ps, respectively.

#### 3.1.2. Limited Growth and Closure of Voids

For model L1, when the impact velocity increased to 0.5 km/s, the preset void was impacted to collapse and became the beginning of damage with limited growth and closure. It can be seen from the stress profile that the evolution period, in this case, was still about 17 ps (Figure 4a). The void was a complete sphere in the initial state, then it collapsed under the shock compression at 1 ps, and finally disappeared at 2 ps, as shown by the construct surface mesh method (Figure 4c). Then, the dislocations were emitted around the void during compression (identified by compression 1), as shown in the microscopic image at 4 ps. It is worth mentioning that the shock wave reached the free surface at this moment, and the peak stress was about 9 GPa (Figure 4a). Following the shock wave evolution, the material reached a tensile state (stretching 1), and at 10 ps, the dislocations decreased, and the damage focused on the preset void position. The porosity increased gradually with the evolution of tension. After 16.5 ps, L1 was compressed again (compression 2), and the evolution of the microstructure repeated the process of dislocation growth from the void position as well as the subsequent deformation around the preset void. For the stretching 2 process, characterized by 29 ps, the microstructure characteristics showed the same pattern as the previous period of stretching 1. Due to the existence of plastic deformation, the temperature rise in the damaged area was more pronounced (Figure 4b). The temperature profile showed a significant temperature rise in the preset void area, and the maximum temperature was close to 450 K due to the evolution of defects mainly concentrated in the preset void position. In the evolution time of 25 ps, the temperature fluctuated in the range of 250–450 K, although it still did not exceed the melting point.

In addition, we calculated the sample volume and void volume fraction in several evolution periods and found that the volume fraction increases first and then decreases in each period, proving the limited growth and closure of voids (Figure 5a). Comparing different evolution periods, it can be found that changes of the sample volume gradually decrease over a long evolution period, and the void volume fraction also decreases, indicating that the damage is less obvious. It should be noted that the collapse of the void also affects the crystal structure. Figure 5b shows that a large number of disordered lattice structures are concentrated around the collapsed void.

At the current impact velocity, similar to L1, the preset void collapsed first (at 1 ps) in L2, L3, and L4, with a complete collapse and dislocations emission at 2 ps. Then, the dislocations extended in the confined body with nucleation characterized by single-point nucleation from the preset void. However, there were some changes under the current shock pressure with a decrease in the strain rate. First, in the process of dislocation propagation, the lower the strain rate, the wider the propagation range. In particular, the dislocations of L1 were mainly concentrated near the center (Figure 6a). Second, the spall phenomenon also showed a specific law that decreasing the strain rate eliminates the shrinkage characteristics of the defect area and causes the material to spall directly, which is due to the temporal competition between voids growth and the shock waves release. In other words, a decrease in strain rate is conducive to the growth of preset void and the formation of larger voids.

The stress profile after the occurrence of damage shows that as the strain rate decreases, the stress exhibits a more pronounced strain localization, and models with higher strain rate show compressive stress at the maximum damage time (Figure 7a). After spall, the temperature near the spall surface exceeds the melting point, proving the material’s local melting (Figure 7b). The sample volume, void fraction, and void number were also compared (Figure 7c,d). It can be found that with a decrease in the strain rate, the periodic changes in the sample volume are less obvious, explaining the occurrence of plastic deformation. In model L2, the strain rate is relatively high, and the void still maintains the characteristics of limited growth and closure, which gradually tends to achieve a balance with the evolution period. The number of voids tends to decrease in each individual period, indicating the aggregation and coalescence of voids. For samples with a low strain rate, it is difficult to observe the peak value and resilience of void volume, suggesting direct material spall.

#### 3.1.3. Typical Spall Damage

With the further increase of impact velocity to 1 km/s, the periodic phenomenon of elastic deformation is less significant and samples show typical spall. For model L1, the void collapses at 0.5 ps and emits dislocations at 1 ps (Figure 8a). The peak stress is nearly 20 GPa at 3.5 ps when the shock wave reaches the free surface. After 1 ps, the dislocations develop. It should be noted that at 8 ps, multiple nucleation occurs at the void position and the corresponding plane, and the number of dislocations in other positions decreases. Finally, at 15 ps, the obvious spall occurs. Compared with that of v_0_ = 0.5 km/s, the damages are located throughout the spall plane, being no longer localized near the preset void. Comparing samples with other strain rates, the collapse process of the preset void is faster than that of 0.5 km/s, and the propagation range of dislocations extends to the whole material for all models. Moreover, the samples show the characteristics of multipoint nucleation. However, the nucleation position is slightly different, that is, when the strain rate is low, it no longer concentrates near the preset void position, and a wider range of damage is formed as the strain rate decreases, obtained from the microstructure when obvious damage happened (Figure 8).

Similar to model L1 with an impact velocity of 1 km/s, the void collapses at 0.5 ps when the impact velocity reaches 2 km/s (Figure 9a). The difference is that due to the high impact velocity, the microscopic image shows that the area where the shock wave passes through is severely damaged. In addition, because of the considerable damage, there is barely no dislocation in this case. When the evolution comes to 3 ps, the shock wave reaches the free surface with a corresponding peak stress of about 45 GPa (Figure 10a). The damage continues to evolve, with multiple nucleation at 7 ps and fracture at 10 ps. The same phenomenon can be observed for samples with other strain rates, and the degree of atomic disorder is greater than before. However, when decreasing the strain rate, few dislocations will occur in the tensile process, and the extension range becomes wider as the strain rate decreases. The nucleation process is characterized by uniform nucleation for all samples, and the preset void has almost no effect on it. Noting that the difference here is that under relatively weak shocks, the extension of dislocations appears first, and the nucleation starts at the intersection of dislocations rather than direct nucleating. Similarly, with the decrease of the strain rate, the damaged area becomes wider seen from the microstructure images with obvious damage.

In order to observe whether samples melt at such spall, the temperature profile was analyzed at an impact velocity of 2 km/s. It can be seen that the temperature of L1 basically reaches the melting point when the shock wave reaches the free surface (Figure 10b). When spall developed to 10 ps, the temperature at the void position is significantly higher than that at other positions, reaching at 1500 K, which is higher than the melting temperature of aluminum, indicating that melting has occurred around this area. Comparing the pressure and temperature profile of the different samples after damage, it can be found that the spall region forms an uneven stress distribution and temperature rise simultaneously due to the occurrence of spall (Figure 10a,b).

In particular, the sample volume as well as damage length were counted at a 2 km/s impact velocity (Figure 10c,d). The conclusion can be drawn that sample volume increase over time, revealing the occurrence of typical spall, and the damage length increases rapidly with decreasing the strain rate.

### 3.2. Spall Strength

Spall strength is a crucial criterion to evaluate the mechanical properties of materials [25,26]. In general, the spall strength is characterized by the free surface velocity in the experiment: σ = 0.5ρ_0_cΔu, where ρ_0_ is the initial density and c = 6.305 km/s is the sound speed along the [010] direction of an ideal aluminum EAM crystal. Δu is the pullback amplitude from the free surface velocity profiles. Here, the free surface velocities of all examples are summarized, as shown in Figure 11. For comparison, aluminum without a preset void is also shown (dotted lines). It can be seen that there is no apparent tensile stress relaxation (spall pulse) and pullback signal under low-pressure shock (0.25 km/s) regardless of whether there is a preset void or not. When the impact velocity is 0.5 km/s, the samples without a preset void will only experience elastic deformation and no spall signals. However, the samples with a preset void, especially at low strain rate, have obvious pullback signals. The occurrence of spall also shows that the existence of the preset void significantly reduces the strength of the sample in this case. When the velocities are increased to 1 and 2 km/s, the pullback signals are obvious due to the occurrence of spall. In this case, the influence of the preset void on the free surface velocity is no more obvious.

The calculated spall strengths of all damaged samples were presented (Figure 12), and the effects of strain rate on spall strength were also compared. It can be concluded that, on the whole, the spall strength increases with increasing the strain rate at all impact velocities. In addition, the strain rate hardening effect has recently been discussed in previous studies [11,27].

In particular, when the shock pressure is low (impact velocity of 0.5 km/s), the samples with a preset void are more prone to spall and the samples without preset void exhibit no spall, indicating that the existence of a preset void can effectively reduce the spall strength, as expected. At the impact velocity of 1 km/s, it can be seen that the spallation strength without preset voids is significantly higher than that with preset voids. However, at a higher shock pressure, the influence of a preset void on spall strength is no longer obvious. At this point, the difference of spall strength of the sample mainly depends on the difference of strain, and it seems that the strain rate effect of high loading strength (2 km/s) is weaker than that of relatively low loading strength (1 km/s).

## 4. Conclusions

In this work, we studied the effects of preset void on the microstructural evolution and spall damage of aluminum under shock compression and release with the molecular dynamics method. The response characteristics of the preset void under different shock pressures and release strain rates were revealed.

During the elastic deformation, the preset void oscillates with multiple compressions and tensions, while the oscillation gradually disappears with the increase of the impact velocity. The preset void can be the beginning of plastic deformation and contribute to the development of damage. Studies also show that the decrease of the release strain rate is conducive to the growth of the preset void, resulting in significant strain localization and the occurrence of fracture. The damage can be repaired after a certain period at a high strain rate. As the shock pressure increases to a certain level, the preset void collapses directly after compression and the lower the strain rate, the wider the damaged area, characterized by the homogeneous nucleation. More importantly, at the low impact velocity (<1 km/s), the existence of preset void can effectively reduce the spall strength and strength of spallation are determined by both the preset defects and strain rate. However, under strong shock (>2 km/s), the difference of the spall strength of the sample mainly depends on the difference of strain rate.

## Figures and Tables

**Figure 1 nanomaterials-12-01853-f001:**
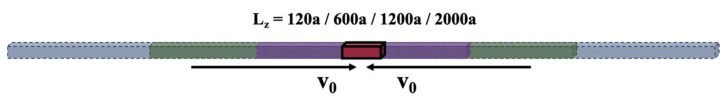
Initial configuration of the simulated sample with the preset void in the center. The compression process was generated by the impact between the two parts of the sample.

**Figure 2 nanomaterials-12-01853-f002:**
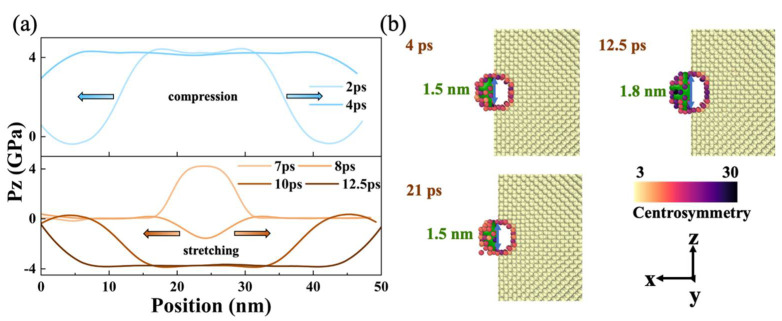
(**a**) Evolution of longitudinal stress profile at v_0_ = 0.25 km/s, where Pz is the stress component along the z direction. Compression and stretching processes are represented by different colors. At the beginning of the impact (2, 4 ps), compression occurs, and the pressure is positive. After 7 ps, it changes to the tensile process, and the stress is negative. (**b**) Microscopic views of void evolution changes at 4, 12.5, and 21 ps. Left: atoms with centrosymmetry values of less than 8 were deleted. Right: the plane shown here was 1 nm thick at the center of the void.

**Figure 3 nanomaterials-12-01853-f003:**
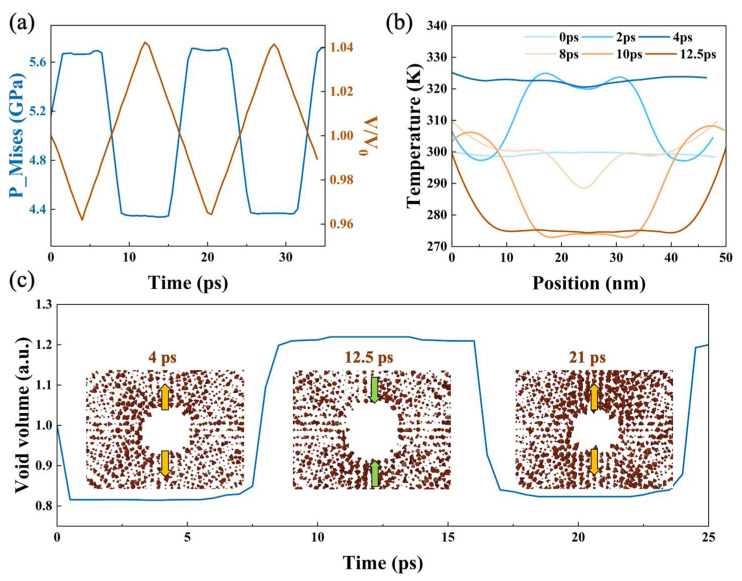
(**a**) Sample volume change and the von Mises stress of the preset void layer, where the blue curve represents von Mises stress and the brown curve represents sample volume. Horizontal axis is along z direction of the sample. (**b**) Evolution of temperature profile at v_0_ = 0.25 km/s, and different colors correspond to different evolution times. (**c**) Changes in the preset void volume and microscopic images present atomic displacement vectors.

**Figure 4 nanomaterials-12-01853-f004:**
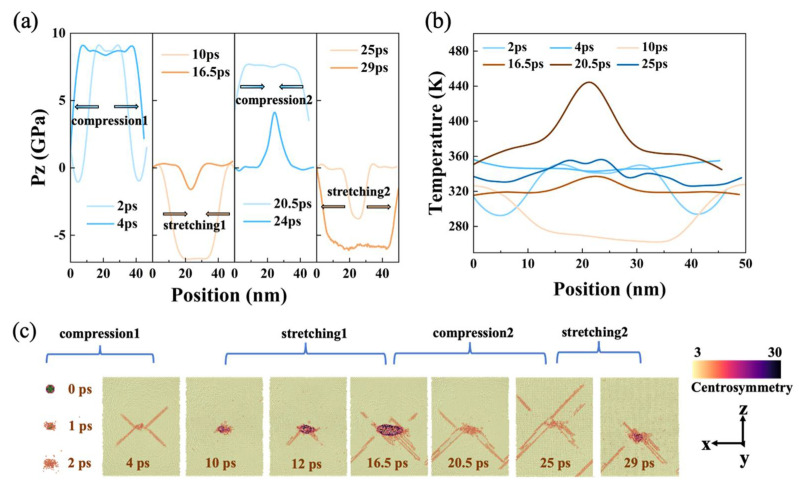
Evolution of (**a**) longitudinal stress profile and (**b**) temperature profile at v_0_ = 0.5 km/s. (**c**) Microscopic views of void evolution at 0, 1, and 2 ps (atoms with centrosymmetry values less than 8 were deleted), 4, 10, 12, 16.5, 20.5, 25, and 29 ps (the microscopic views were sliced from the center axis).

**Figure 5 nanomaterials-12-01853-f005:**
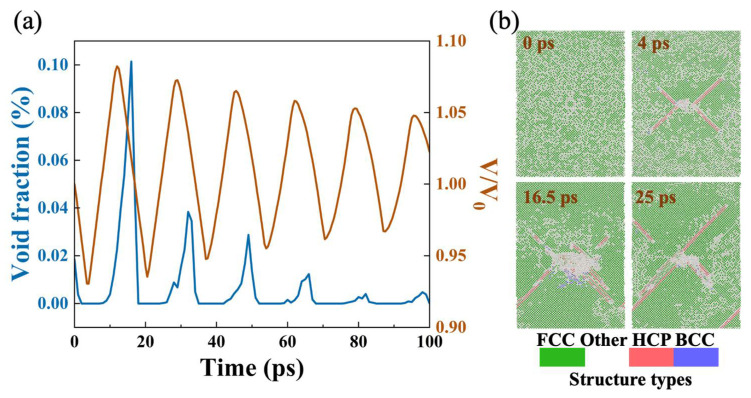
(**a**) Changes in sample volume and void volume fraction. (**b**) Types of structure around the preset void at 0, 4, 16.5, and 25 ps.

**Figure 6 nanomaterials-12-01853-f006:**
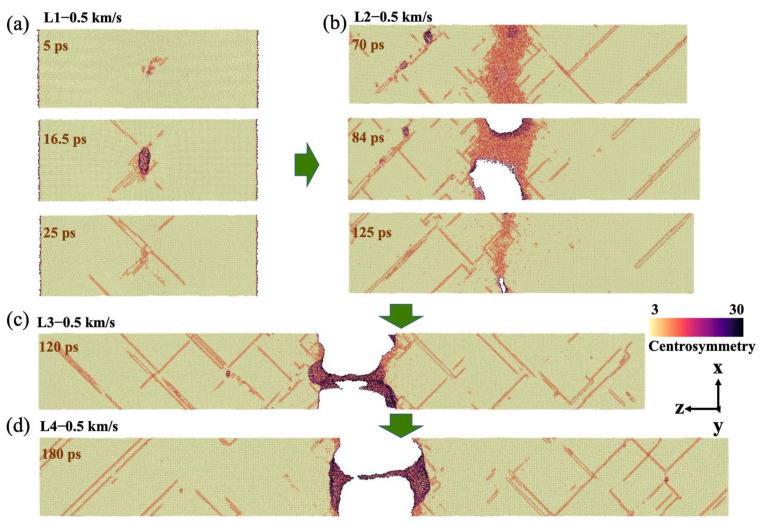
Microscopic views of void evolution of (**a**) L1, (**b**) L2, (**c**) L3, and (**d**) L4 at specific times (microscopic views were sliced from the center axis) at v_0_ = 0.5 km/s.

**Figure 7 nanomaterials-12-01853-f007:**
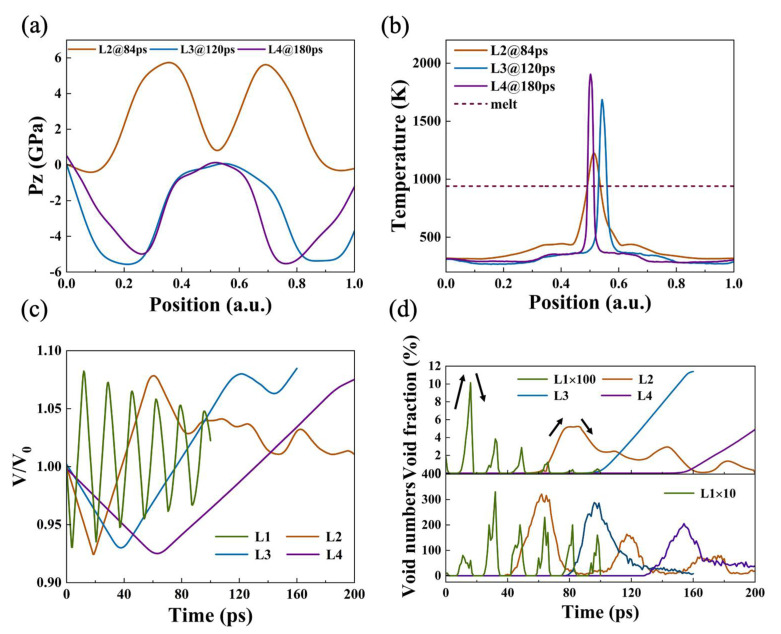
(**a**) Longitudinal stress profile and (**b**) temperature profile of L2, L3, and L4 at 84, 120, and 180 ps, respectively, at v_0_ = 0.5 km/s. The red dotted line represents the melting temperature of 940 K for aluminum under the EAM potential. Evolution of (**c**) sample volume and (**d**) void volume fractions and void numbers of L1, L2, L3, and L4.

**Figure 8 nanomaterials-12-01853-f008:**
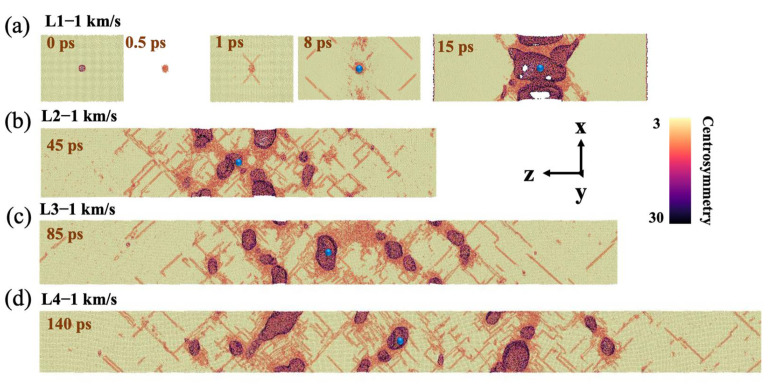
Microscopic views of void evolution and damage for (**a**) L1, (**b**) L2, (**c**) L3, and (**d**) L4 at v_0_ = 1 km/s. The microscopic views were sliced from the center axis, and atoms with centrosymmetry values less than 8 were deleted. The blue ball represents the preset void position.

**Figure 9 nanomaterials-12-01853-f009:**
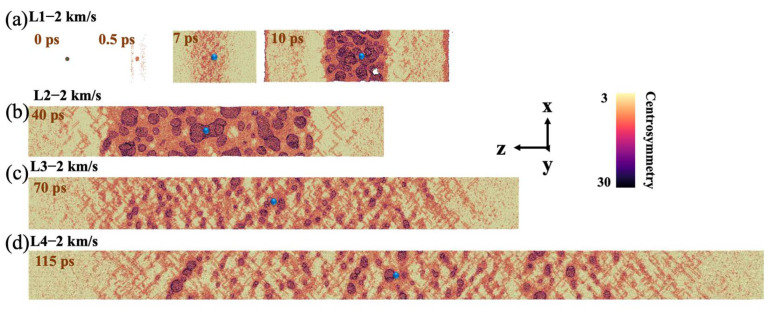
Microscopic views of void evolution and damage for (**a**) L1, (**b**) L2, (**c**) L3, and (**d**) L4 at v_0_ = 2 km/s. The microscopic views were sliced from the center axis, and atoms with centrosymmetry values less than 8 were deleted.

**Figure 10 nanomaterials-12-01853-f010:**
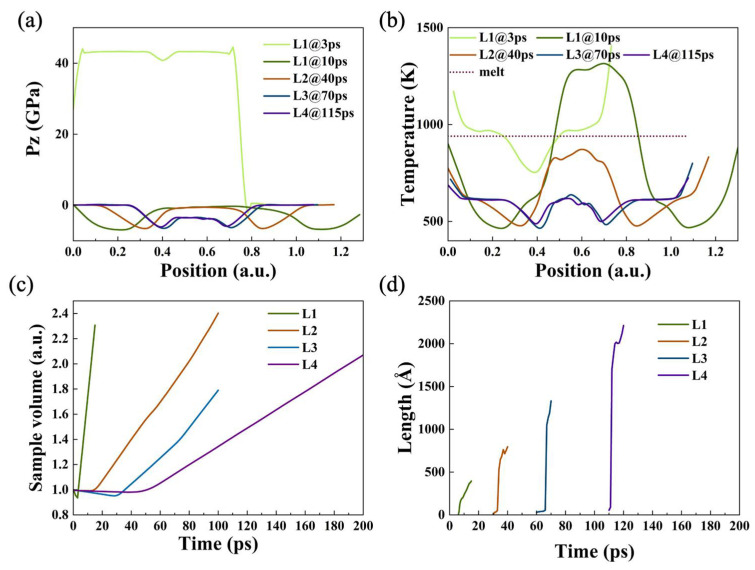
(**a**) Longitudinal stress profile and (**b**) temperature profile of L1 (3, 10 ps), L2 (40 ps), L3 (70 ps), and L4 (115 ps) at v_0_ = 2 km/s. Evolution of (**c**) sample volume, and (**d**) damaged area width with time at v_0_ = 2 km/s.

**Figure 11 nanomaterials-12-01853-f011:**
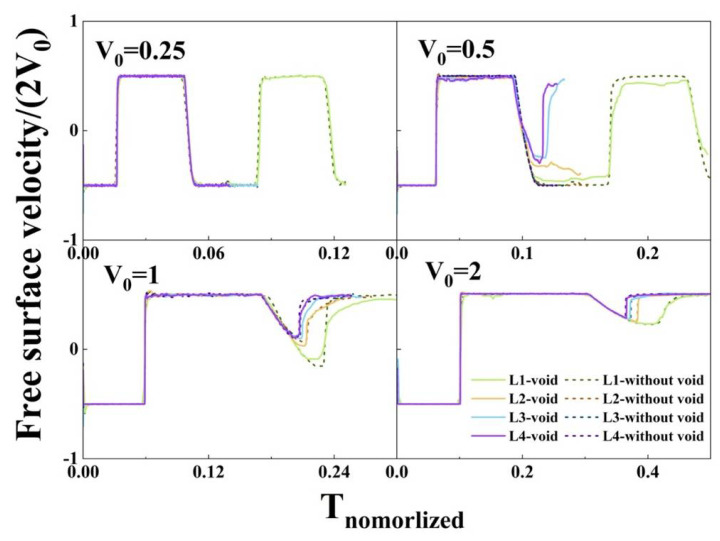
Free surface velocity profile of the samples. Both axes are normalized. T_normalized_ = t × 2 v_0_/L_z_.

**Figure 12 nanomaterials-12-01853-f012:**
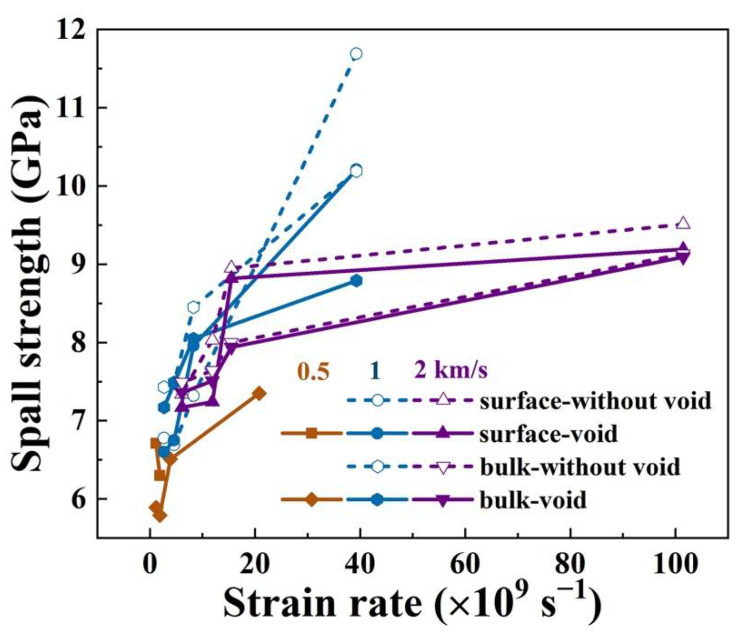
Spall strength of samples at different impact velocities and strain rates, the dotted line represents single-crystal aluminum without a preset void, and different colors represent different impact velocities. The spallation strength from the free surface velocity approximation and directly obtained from the stress profiles is given.

**Table 1 nanomaterials-12-01853-t001:** Release strain rates (×10^9^ s^−1^) of all samples at different impact velocities.

Sample Size	Impact Velocity (km/s)			
	0.25	0.5	1	2
L1	10.02	20.78	39.32	101.51
L2	1.86	3.85	8.34	15.50
L3	1.13	1.92	4.57	11.98
L4	0.60	1.23	2.70	6.04

## Data Availability

The data presented in this study are available on request from the corresponding author.

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
