# Peer review of "Evolution of Preset Void and Damage Characteristics in Aluminum during Shock Compression and Release"

_nanomaterials, 2022, doi:10.3390/nano12111853_

Round 1

Reviewer 1 Report

The authors use the open source package LAMMPS [17] to study the mechanical properties of shock compression of aluminum  structures FCC, HCP and BCC. The von Mises stress equation is chosen as a criterion od damage. Change of the sample volume in time is presented in Figures 3 and 5. Section 3.1.3 is devoted to a typical spall damage.

I have the following general remark.

I note that the authors declare general results based on their particular simulations. I mean the phrase on line 104 “In order to analyze the simulation results comprehensively and accurately…” and others. The present results are rather observations, what do we see, than comprehensive and accurate investigation. The study of waves and oscillations might require a Fourier analysis. The damage investigation requires  a wide statistical analysis. These points are absent in the paper.

Therefore, I suppose that the authors should perform comprehensive and accurate analysis of their simulations or to frankly explain that it is just a numerical experiment, maybe not accurate.

Figure  3c displays three discrete pictures in time. The authors say about the “continuous change of the sample volume”. Is it possible to prepare an animation to really demonstrate a continuous change?

Author Response

Reviewer #1:

The authors use the open source package LAMMPS [17] to study the mechanical properties of shock compression of aluminum structures FCC, HCP and BCC. The von Mises stress equation is chosen as a criterion od damage. Change of the sample volume in time is presented in Figures 3 and 5. Section 3.1.3 is devoted to a typical spall damage.

I have the following general remark.

I note that the authors declare general results based on their particular simulations. I mean the phrase on line 104 “In order to analyze the simulation results comprehensively and accurately…” and others. The present results are rather observations, what do we see, than comprehensive and accurate investigation. The study of waves and oscillations might require a Fourier analysis. The damage investigation requires a wide statistical analysis. These points are absent in the paper. Therefore, I suppose that the authors should perform comprehensive and accurate analysis of their simulations or to frankly explain that it is just a numerical experiment, maybe not accurate.

Response: Thank the reviewer for pointing out these details. As you said, a comprehensive analysis of our simulations is needed, to clarify the detailed interaction mechanism between shock wave and microscopic defects in materials. In fact, this paper focuses on the material damage properties under different impact pressures and strain rates. Based on MD simulations, we found that the oscillation and complete collapse of void, and local plastic deformation and damage growth under different shock intensities. And the reduction of strain rate can increase the development process of voids. The comprehensive and accurate investigation is inappropriate and can cause misunderstanding, so the relevant parts have been changed in the manuscript.

Figure 3c displays three discrete pictures in time. The authors say about the “continuous change of the sample volume”. Is it possible to prepare an animation to really demonstrate a continuous change?

Response: Figure 3c shows the volume evolution of corresponding void during the periodic change of impact compression and release, and relevant animation has been provided for more detailed and continuous time evolution.

Reviewer 2 Report

Please find my comments in the attached pdf file.

Best Regards.

Author Response

Reviewer #2:

I evaluated the manuscript “Evolution of preset void and damage characteristics in aluminum during shock compression and release” by Ya-Ting Wan et al. The manuscript studies the damage characteristics under different shock compression and release with molecular dynamics simulations (LAMMPS). The authors reported on the effects of the increasing of shock pressure or impact velocity. Furthermore, the effect of preset void on the spall strength of the material was discussed. The manuscript is well written and the introduction is clear. I think that the manuscript may be accepted for publication on MDPI Nanomaterials provided that the authors address the following points:

Fig 2a: The caption of Fig 2a should be expanded in order to describe better the several traces appearing in the panel. Why do the authors call the stress “P_z”? Is the stress along z direction? The authors should comment on that.

Response: Thank you for the patient reading. For better understanding, the caption of Fig 2a has been revised to “Evolution of longitudinal stress profile at v0 = 0.25 km/s, where Pz is the stress component along z direction. Compression and stretching processes are represented by different colors. At the beginning of the impact (2, 4 ps), compression happens, and the pressure is positive. After 7 ps, it goes to the tensile process, and the stress is negative.”

Page 3: lines 108-110: “In particular, the deformation of each atom was analyzed by its centrosymmetry parameter, which is typically used to measure the local disorder around a single atom and identify whether 110 the atom is in the ideal lattice or in the defect.” The authors should comment more on the centrosymmetry parameter and define it quantitatively, since it is plotted in some figures.

Response: The centrosymmetry parameter is a useful measure of the local lattice disorder around an atom and can be used to characterize whether the atom is part of a perfect lattice, a local defect (e.g. a dislocation or stacking fault), or at a surface, which is defined as follows:

Ri and Ri+6 are the vectors corresponding to the six pairs of opposite nearest neighbors in the initial fcc lattice, and more detailed information can be found on the official website of Ovito software.

The authors should comment more in the text why the temperature is inhomogeneous in the system (see Fig. 3b) and why it changes with time. What is the thermal problem of the system?

Response: Thank the reviewer for pointing out this key issue. Before impact, the temperature of the system is stable at a uniform 300 K. After the compression, with the generation of shock wave, the temperature will rise and propagate in the sample like the stress profile. Because of symmetrical compression, the temperature waveforms on both sides of the sample are basically symmetrical. As for the change of temperature with time, it is related to the evolution of microscopic structure induced by impact. Relevant detailed explanations have been added in the article.

Fig 3: The caption does not describe well the traces appearing in panel a and b. The authors should add some comments. The combination of colours reported in panel a is not optimal, because green and brown are similar. The authors should change the colours. One possibility could be the use of blue instead of green (such as Fig 5a), but I leave this choice to the authors. How is the position (horizontal axis of panel b) defined? Along which direction in the crystal is the position defined?

Response: The caption of Fig. 3a-b have been modified as “(a) Sample volume change and von Mises stress of the preset void layer, where the blue curve represents von Mises stress and the brown curve represents sample volume. Horizontal axis is along z direction of the sample. (b) Evolution of longitudinal temperature profile at v0 = 0.25 km/s, and different colors represent the sample temperature at different evolution times.”. The colours of Fig. 3a have been optimize as well.

The authors could mention an alternative approach to evaluate the quality and the damages present in real samples. For instance, the application of pump-probe measurements together with cluster techniques may classify the quality of the samples depending on the tiny differences of the optical readout upon the pumpprobe measurements. I mention the following two manuscripts as possible starting points: S. Peli et al., https://doi.org/10.1038/s41598-020-72534-1 A. Ronchi et al., https://doi.org/10.1016/j.ultras.2021.106403

Even if this paper is theoretical and the latter technique is experimental, I believe that it would be worth to mention it in the introduction (or conclusions), giving a few references as well, to increase the impact and the audience of the paper.

Response: The alternative approach to evaluate the quality and the images present in real samples and relevant references have been supplemented in the introduction section. Thank the reviewer again for valuable comments.

Round 2

Reviewer 1 Report

If the animation will be included in the paper, the supplementary file shoud be described. A description of numerical data of the animation shoud be also done in the bulk paper or in supplementary file.  

I think that the paper could be published in the present form.

Author Response

Reviewer #1:

If the animation will be included in the paper, the supplementary file should be described. A description of numerical data of the animation should be also done in the bulk paper or in supplementary file.  

I think that the paper could be published in the present form.

Response: Thank you for the patient reading. The description of the animation has been added in the appropriate position in the manuscript.

Reviewer 2 Report

Please find attached a pdf file with my comments.

Best Regards.

Author Response

Reviewer #2:

I evaluated the improved version of the manuscript “Evolution of preset void and damage characteristics in aluminum during shock compression and release” by Ya-Ting Wan et al. I believe that the manuscript has improved and now deserves publication.
I have just one minor comment:

By reading the new version of the introduction I think that mentioning the cluster techniques improves the impact of the paper. However, I realize that maybe some authors cannot be familiar with the pump-probe technique. For this reason, maybe some other references on this technique may be added. Here I put a couple as a starting point:

doi.org/10.1103/PhysRevB.85.195431 doi.org/10.1021/acs.jpcc.2c01060 Thank you very much for your attention. Best Regards.

Response: Thank the reviewer for valuable comments and relevant references have been added for better understanding.